CellPress

## Perspective

# Three decades of ethical, legal, and social implications research: Looking back to chart a path forward

Deanne Dunbar Dolan,[1,*] Sandra Soo-Jin Lee,[2] and Mildred K. Cho[3]

[1]Center for ELSI Resources and Analysis (CERA), Stanford Center for Biomedical Ethics, Stanford University School of Medicine, Palo Alto, CA 94305, USA
[2]Division of Ethics, Department of Medical Humanities & Ethics, Columbia University, New York, NY 10032, USA
[3]Departments of Medicine and Pediatrics, Stanford Center for Biomedical Ethics, Stanford University School of Medicine, Palo Alto, CA 94305, USA
*Correspondence: ddolan@stanford.edu

## SUMMARY

More than thirty years ago in the United States, the National Center for Human Genome Research (NCHGR) at the National Institutes of Health (NIH) and its partner in the Human Genome Project (HGP), the Department of Energy (DOE), called for proposals from social scientists, ethicists, lawyers, and others to explore the ethical, legal, and social implications (ELSI) of mapping and sequencing the human genome. Today, nearly twenty years after the completion of the HGP, the ELSI Research Program of the National Human Genome Research Institute (NHGRI) continues this support. It has fostered the growth of ELSI research into a global field of study, uniquely positioned at the nexus of many academic disciplines and in proximity to basic and applied scientific research. We examine the formation of the first ELSI program and consider whether science policy in the public interest can exist within the confines of a set-aside from the NHGRI budget.

## INTRODUCTION

In the U.S., "ELSI" refers to the field of study concerned with the ethical, legal, and social implications of genetics and genomics (Table 1). Between its origins in the international Human Genome Project (HGP) (which sequenced and mapped the complete human genome from 1990 to 2003) and today, ELSI scholars have produced thousands of articles, books, and other materials.[1] These works explore a variety of issues in basic research and its clinical translation, as well as broader societal issues raised by emerging technologies in the life sciences.[2] Globally, ELSI researchers are positioned in proximity to, or embedded in, large life science initiatives; focus on the anticipation of, or rapid response to, emerging scientific issues; support the co-design of research agendas with the public; interact with a broad range of stakeholders (the media, policy makers, and industry); and use diverse source materials and approaches (see Table 2).[3,4] Although heterogeneous in its methods, it is our contention that ELSI inquiry is focused on distinct objects for unique purposes compared with adjacent fields, such as bioethics, and nearby disciplines.

ELSI researchers identify and explore issues associated with the conduct and application of scientific research.[15] There are many reasons to take up this study, including facilitating public discussion and ensuring that these explorations are conducted for the benefit of society, their benefits are evenly distributed, and harms and misuse are limited to the greatest extent

possible.[21] These critical missions are not necessarily oppositional to scientific discovery. As their work is in the public interest, ELSI researchers strive for intellectual independence from both the science they observe and institutional or financial pressures that conflict with their mission. We proceed from the premise that although smaller sources have funded U.S. ELSI work and the "ELSI community" is enriched by scholars without grant-based funding, substantial and sustained funding by the ELSI Research Program at the National Human Genome Research Institute (NHGRI) has been, and remains, a primary support to the field of study.[1,22–24] In this paper we consider "the ELSI hypothesis"—the idea that the best way to produce informed science policy for the HGP would be a formal budget "set-aside" within it—and offer some lessons to inform the future direction of the ongoing U.S. federal funding program.

## ESTABLISHING ELSI RESEARCH IN THE UNITED STATES

In 1990, the international effort to map and sequence the human genome promised to dramatically enhance our understanding of human biology and of both genetic and acquired diseases.[25] However, in light of atrocities inspired by eugenics movements, which in America had provided the rationale for state-sponsored mass sterilization programs and in Europe had inspired the Holocaust perpetrated by Nazi Germany, it was clear that the ability to ascertain genetic information would bring with it the possibility

**Table 1. Acronym definitions**

| | |
|---|---|
| BERAC | Biological and Environmental Research Advisory Committee (Department of Energy) |
| CEER | Center of Excellence in ELSI Research |
| CERA | Center for ELSI Resources and Analysis |
| DHHS | Department of Health and Human Services |
| DOE | Department of Energy |
| ELSI | Ethical, legal, and social implications |
| ERA | ELSI Research Advisors |
| ERPEG | ELSI Research Planning and Evaluation Group |
| GSWG | Genomics and Society Working Group |
| HGP | Human Genome Project |
| NACHGR | National Advisory Council for Human Genome Research |
| NIH | National Institutes of Health |
| NCHGR | National Center for Human Genome Research |
| NHGRI | National Human Genome Research Institute (formerly NCHGR) |
| OHER | Office of Health and Environmental Research, Office of Energy Research, Department of Energy |
| OPCE | Office of Policy, Communications, and Education (NIH) |
| RAC | Recombinant DNA Advisory Committee |

of stigma and discrimination for carriers of genetic disorders, inhibit equitable access to U.S. health care, and even change the composition of human society, if genetic information were used to shape reproductive planning.[26–29] The architects of the HGP could not ignore the cultural currents of the preceding decades, including the nature or nurture debates in psychology, the rise of global human rights discourses, or the evolution of medical ethics and human subjects protections.[30] In fact, both the National Academy of Sciences and the Congressional Office of Technology Assessment had discussed ethical and social issues in the planning and feasibility reports for the HGP.[31,32]

Speaking to these issues, James Watson, co-discoverer of the molecular structure of DNA and the first HGP director at the National Institutes of Health (NIH), announced at an October 1988 press conference that a portion of the project budget would be used to study its impacts on society.[33,34] The first action in this direction was the establishment of a working group on ethics in January 1989 to coordinate efforts at the NIH and the Department of Energy (DOE), the two primary U.S. HGP agencies (for the DOE interest, see Annas and Elias[35]).[36] In March 1989, the Office of Human Genome Research (OHGR) added a program announcement to the NIH Guide to Grants and Contracts to request the first applications for ELSI research (see Box 1).[36]

The working group on ethics held its first formal meeting on September 14 and 15, 1989, one month before the Department of Health and Human Services (DHHS) established the National Center for Human Genome Research (NCHGR) at the NIH.[36,37] This meeting resulted in a mission statement that inspired the NCHGR to establish an ELSI branch within its Division of Extramural Research and the DOE to start an ELSI program in the Office of Health and Environmental Research (OHER) and a revised NIH program announcement.[36,38] ELSI grant applications from members of the academic research community were now encouraged in nine topic areas (see Box 2). The parallel DOE

grants program was preferentially focused on privacy, fair use of genetic information, and education of the public.[33,36,39]

In November 1990, the two agencies announced the formation of the Joint Working Group on Ethical, Legal, and Social Issues (the "ELSI working group") to steer the course of the two extramural research programs, convene various task forces and conferences, and coordinate the production of policy options.[39,40] It was initially chaired by Nancy S. Wexler, PhD, a clinical psychologist, and comprised experts in law, ethics, genetics, clinical medicine, and other fields.[41] During its tenure, the group established task forces on privacy and insurance and directed the distribution of grant funds for research on access to high-quality genetic tests, the fair use of genetic information by employers and insurers, privacy issues, and public and professional education.[29,42,43] By September 1991, the NCHGR had funded twenty-five extramural ELSI grants and ten national conferences, including a January 1991 workshop that resulted in a social policy research agenda that is of continued relevance today (see Box 3).[35,39,40]

ELSI research grants received 3% of the annual budget of the NCHGR, with a budgeted scale up to 5% within the first three years (this occurred in 1991) and a consistent 3% of the OHER/DOE budget.[33,34,43] In addition to the funds that were earmarked for ELSI research grants, the activities of the ELSI working group were jointly funded from the administrative budgets of the NCHGR and DOE.[44] The financial commitment by the DOE was the idea of Tennessee Democratic senator Al Gore, who challenged a DOE official in a 1989 subcommittee hearing, upon learning that the agency had not created an ELSI set-aside: "I think that whether you set aside the money or not will be a signal of whether you are really taken seriously or not. I would like to see not so much a duplicate of the NIH effort as I would like to see in the joint plan you develop a comparable commitment of money."[45] The total sum of this first-of-its-kind ethics

**Table 2. Examples of global ELSI initiatives, 1990 to present**

| Location | Date | Initiative(s) |
|---|---|---|
| Austria, Finland, and Germany | 2009 | Austria, Finland, and Germany launched a multinational initiative called ELSAGEN to fund collaborative research on ELSA issues associated with genomics and the related sciences.[5,6] |
| Canada | 2000 to present | Genome Canada, a not-for-profit corporation partly funded by the Federal Government of Canada, funds the Genomics, Economic, Ethical, Environmental, Legal and Social Aspects (GE$^3$LS) program.[5,7] |
| | 2001–2011 | The Canadian Institutes of Health Research, Institute of Genetics, includes the study of genetics and the ethical, legal, and social issues it raises as a strategic research priority.[5] |
| | 1992–1997 | The Medical, Ethical, Legal and Social Implications (MELSI) of genetics program was a component of the Canadian Genome Analysis and Technology (CGAT) initiative, the Canadian contribution to the HGP.[7,8] |
| European Union | 2013–2020 | The European Commission made responsible research and innovation (RRI) a cross-cutting theme in the Framework Program for Research and Innovation, Horizon 2020, and assigned responsibility for RRI to the Science with and for Society (SwafS) sub-program. RRI themes include public engagement, open access, gender, ethics, and science education.[9,10] |
| | 2002–2012 | The Economic and Social Research Council funded centers and institutions across the United Kingdom (Cesagen, Innogen, Egenis, and Genomics Forum) to study the economic and social implications of genomic science and technologies. Together, these centers were called the Economic and Social Research Council (ESRC) Genomics Network or EGN.[5,11] |
| | 1994–1998 | The 4th European Union Framework Program introduced ELSA as a label for funding research into the ethical, legal, and social aspects of emerging sciences and technologies, stakeholder dialogues, education, and other activities.[3,12] |
| The Netherlands | No date available | The Societal Component of Genomics (MCG) program of the Dutch Research Council or Netherlands Organization for Scientific Research (NWO) funded researchers in the social sciences and humanities to anticipate developments in science and society.[6,3,13] |
| | 2009–2011 | The Dutch government organized the Committee Societal Dialogue Nanotechnology (CieMDN) and tasked it with organizing a national public dialogue on nanotechnology called Dutch Nanodialogue that was active from March 2009 to January 2011.[14] |
| | 2005–2010 | The ELSA (ethical, legal, and societal aspects) coordinating project, Societal Aspects of Genomics of the Sixth Framework Program project, ERA-SAGE, was coordinated by the Netherlands Organization for Scientific Research beginning in 2005. It coordinated ELSA activities in eight national funding agencies (the Netherlands, United Kingdom, Austria, Norway, Finland, Germany [2], and Canada) and three funding agencies with a specific interest in this field (Israel [2] and Switzerland). It has been argued that ELSA in these countries expanded beyond genomics in 2005 and began to be applied to other emerging technologies, such as nanotechnology, information and communication technologies, synthetic biology, and neurotechnology.[15,16] |
| | 2004 | The Netherlands Genomics Initiative (NGI) allocated 5% of its budget to two initiatives: (1) researcher-driven projects on "the societal component of genomics research" and (2) the Centre for Society and Genomics (CSG) (later renamed the Centre for the Study of Life Sciences), which housed ∼50 ELSA research projects.[3,4,5,6] |
| | 2001 | The Dutch government allocated €189 million to genomics research and earmarked 4% for the study of ethical, social, economic, psychological, and legal aspects of the genomics programs and the establishment of the Netherlands Genomics Initiative (Nationaal Regie-Orgaan Genomics), an independent task force charged with governing the new genomics infrastructure.[17] |

(*Continued on next page*)

**Table 2.** *Continued*

| Location | Date | Initiative(s) |
|---|---|---|
| Norway | 2014 to present | In addition to the Research Council of Norway (RCN) ELSA I and ELSA II programs, early Norwegian national biotechnology programs, such as those for functional genomics (FUGE) and nano materials (NANOMAT), include ELSA research components. Ongoing programs, such as BIOTEK2021 and NANO2021, allocate 2%–5% of their funding to ELSA.[18] |
| | 2008–2014 | The ELSA II program period was focused on nanotechnology and new materials.[15,18] |
| | 2002–2007 | The ELSA of Nanotechnology, Biotechnology and Neurotechnology Program was established by the Research Council of Norway (RCN) to study issues associated with biotechnology, nanotechnology, and cognitive science. The first program period (ELSA 1) focused on functional genomics.[5,6,15,18] |
| South Korea | 2001 | The South Korean government funded an ethical, legal, and social implications program.[6,19,20] |

set-aside within a major U.S. scientific initiative inspired Dr. Wexler to describe the ELSI program as the "the largest biomedical ethics program in this country and probably in the world."[34,46,47]

In June 1993, Congress formally instantiated the activities of the ELSI branch at the NIH with legislation called the National Institutes of Health Revitalization Act. This act mandated that "not less than 5 percent" of the NCHGR budget be allocated to "reviewing and funding proposals to address the ethical and legal issues associated with the genome project (including legal issues regarding patents)."[48] That same year, Francis Collins succeeded Watson as director of the NCHGR, with oversight of the HGP and the ELSI branch. He added two professional staff positions and expanded the group to include representation from lay constituencies, clinical professions, and genome science.[33] In 1995, at the end of Dr. Wexler's 5-year term as chair of the ELSI working group, Lori Andrews, a law professor and legal scholar, was elected to replace her.[49]

## RETHINKING ELSI AT THE NATIONAL INSTITUTES OF HEALTH

Early in 1996, an internal dispute about the purpose and autonomy of the ELSI programs arose when Collins vetoed a plan for an anthology on behavioral genetics. Citing interference by the NCHGR in the budget allocation and position statements of the ELSI working group, Andrews resigned in protest in February 1996. Sociologist Troy Duster, who temporarily replaced her, urged that the working group be made autonomous from NCHGR and highlighted the imbalance between allocations to genome science and ELSI.[12,34,44,49,50] In a context in which the "growing gap between diagnostic information and therapeutic capacity is a time-bomb," Duster opined, "the formula for 95% for the mapping and sequencing versus the 5% for the social consequences seems particularly absurd. What about 50:50?"[44]

Reflecting on her experience, Andrews expressed concerns that Collins had "begun to stack" the ELSI working group by adding a genome scientist as a voting member without nomination or vote by the group.[49] Collins denied that he prevented the working group from expressing opinions but conceded the ne-

cessity of "some limits to its autonomy because it is not a free-standing commission."[44] Following her resignation (and perhaps, as Andrews said, as an investigation into her resignation), Collins and Dr. Ari Patrinos, associate director for health and environmental research at the DOE, commissioned an independent review to evaluate the scope of ELSI activities, the role of external advisers in the ELSI program, and how best to structure input on ELSI issues. They appointed the Committee to Evaluate the Ethical, Legal and Social Implications Program of the Human Genome Project (ELSI evaluation committee) on April 30, 1996.[44,49,51]

The December 1996 report of the ELSI evaluation committee found the ELSI working group to be an integral part of the HGP with a mandate "too broad to be satisfied by any single body" and placement "not commensurate with the more global role of some important policy formulation." The report offered three recommendations: (1) that the NIH implement a process for communication and coordination of the ELSI activities on research ethics in genetic studies within the institutes; (2) that it restructure the existing working group into the ELSI Research Evaluation Committee, which would coordinate ELSI grants and set the research agenda; and (3) that it establish a federally chartered advisory committee on genetics and public policy situated in the Office of the Secretary of the Department of Health and Human Services to "assume the role of identifying issues and formulating policy to ensure integration of new genetic knowledge into health care standards."[51] The remainder of the ELSI working group resigned following the external evaluation, in expectation of the proposed new configuration.[49] Thus, it had ended its advisory role to the NHGRI and DOE ELSI grant-making programs, its task force work, and coordination of policy options by 1997.[52]

Also in 1997, the DHHS elevated the NCHGR to the status of a research institute and renamed it NHGRI.[53] At its February 1997 meeting, the National Advisory Council for Human Genome Research (NACHGR) endorsed all three ELSI evaluation committee recommendations. In July 1997, the NACHGR and the Biological and Environmental Research Advisory Committee (BERAC) at the DOE formally established the ELSI Research

**Box 1. Applicants were asked to address the following questions about research on the human genome in the NIH Guide to Grants and Contracts, 1989**

- What are the concerns to society and to individuals?
- What questions in the areas of ethics and law need to be addressed?
- What can be learned from precedents?
- What are the policy alternatives and the pros and cons of each?
- How can we inform and involve the public?[36]

Planning and Evaluation Group (ERPEG). The ERPEG, chaired by ethicist LeRoy Walters, analyzed the portfolio of ELSI research grants, engaged in a strategic planning process that resulted in the ELSI component of the 1998–2003 HGP strategic plan, and provided expert guidance on both extramural ELSI research portfolios until January 2000.[38]

The ERPEG presented its final report at the February 27, 2000 NACHGR meeting. It noted that the NHGRI had spent more than $58 million on the ELSI program and that the DOE had spent $18.2 million at the end of fiscal year (FY) 1999. The committee recommended that the DOE expand the staffing of its program beyond a single individual, a new joint planning and evaluation group, the recruitment of investigators from underrepresented groups, and activities to promote collaboration between HGP scientists and ELSI researchers, citing the existence of "some in the scientific community who remain indifferent or even hostile to ELSI research."[52] Their portfolio analysis noted that the R01 mechanism may account for the very small number of legal, philosophical, theological, sociological, or economic analyses. They noted highly cited publications, policy recommendations (especially a draft "genetic privacy act" by George Annas of Boston University), and the formation of two successful research consortia, among other program accomplishments.[52,54]

With guidance from the ERPEG, Collins and colleagues identified goals for the ELSI Research Program in the HGP strategic plan for 1998–2003, focused on anticipating the reception and use of the results of the project in clinical care, prevention, nonclinical, and policy settings (see Box 4).[55] The ERPEG recommendation to re-establish a joint DOE/NHGRI ELSI planning group did not come to fruition. However, in 2000, NHGRI formed the ELSI Research Advisors (ERA), a subcommittee of the NACHGR, to advise the council on the grants program and plan for the role of ELSI following the completion of the genome sequencing work.[56] ERA produced a white paper in 2003 that called for enhanced integration between the ELSI extramural grants program and the Office of Policy, Communications, and Education (OPCE) within the NHGRI Office of the Director, among other recommendations. The white paper identified the NHGRI as a "federal agency responsible for funding research" with "no mandate to oversee the development of policies" and suggested a division of labor in which ELSI researchers "develop a body of knowledge" and the OPCE translate their findings to policy makers.[56,57]

In its 2005 report to the NACHGR, the ERA identified three "persistent challenges" that should be addressed for the ELSI Research Program to fulfill its mission: (1) increased integration between ELSI and genome research, (2) more effective translation of ELSI research findings into products that can inform policy, and (3) expansion of the disciplinary and demographic diversity of the ELSI community of researchers. Echoing its white paper and a 1992 report by the House Committee on Government Operations, the report had this to say: "as an extramural research program housed within the Federal Government, ELSI is statutorily not capable of developing or presenting in an effective manner specific policy recommendations to the Nation, the Congress, or the executive branch on the full range of problems presented by the Human Genome Project."[43,57] However, the ERA report implied that the translation of academic research to policy was within the mission of the ELSI Research Program because, among others who might do this work, its grantees were most conversant with ELSI research findings. The report offered that the range of expertise in the ELSI research community should continue to expand and noted that the Centers of Excellence in ELSI Research (CEER) consortium funded beginning in September 2004, with a goal of "translat[ing] ELSI research to safe, effective, and just genetic and genomic policies and practices in research, health, and non-medical settings" would partly address translation challenges.[57]

## RECENT NHGRI ELSI FUNDING PRIORITIES

Like its prior strategic plan, the NHGRI objectives for 2011–2021 emphasized the continued translation of genetic science into the clinical setting, including training clinicians to interpret and use genomic data.[58] The DOE ultimately retired its smaller ELSI grant program. Its NHGRI counterpart continued. The 2012 NHGRI reorganization by its director, Eric Green, located the ELSI extramural research program within the Division of Genomics and Society.[59] Today, planning and priority setting at the division

are directed by the NACHGR, with advice from the Genomics and Society Working Group (GSWG) and periodic strategic planning processes.[38,60] NHGRI still allocates at least 5% of its annual extramural research budget to the ELSI Research Program (see Table 3). This amounted to $1.57 million in fiscal year 1990, $18.9 million in 2016, and about $22 million in 2020.[60,61] In 2020, program funds were allocated between investigator-initiated research (68% of $22 million) and program-initiated research (32%) (J. Boyer, personal communication). This sustained level of financial commitment has made the field of genomics unique among U.S. biosciences.[2]

**Box 2. Research topics in the 1990 revision of the NIH program announcement for those seeking grants to conduct ELSI research**

1. Fairness in insurance, employment, the criminal justice system, education, adoption, the military, and other areas
2. Psychological and societal responses to individual genetic information
3. Privacy and confidentiality, including ownership, control, and consent
4. Genetic counseling in conjunction with testing for prenatal, pre-symptomatic, and polygenic conditions and carrier status; testing in the absence of therapeutic options; and population screening versus testing
5. Genetic information and reproductive decisions
6. Introduction of genetics into medical practice, including standards of care, qualification of medical professionals, and education of patients and the public
7. Historical misuses of genetics, especially eugenics, and relevance to the present
8. Commercialization, including property and intellectual property rights, and accessibility of data and materials
9. Philosophical issues such as definitions of health and disease and questions of determinism and reductionism[36,40]

> **Box 3. Social policy research priorities for the Human Genome Project, 1991**
>
> 1. When and how should new genetic tests be introduced into medical practice?
> 2. How can the confidentiality and privacy of an individual's genetic information be preserved?
> 3. How can genetic discrimination by employers and insurance companies be prevented?
> 4. How might the Human Genome Project affect our concepts of "disease," "normalcy," and "humanness"?
>
> Source: ELSI Priority-Setting Workshop (January 1991), National Center for Human Genome Research, National Institutes of Health, Bethesda, Maryland.[35]

Today, the ELSI Research Program, often in partnership with other NIH institutes or centers, funds U.S. investigators in four overlapping research areas: genomics and sociocultural structures and values, genomics at the institutional and system level, genomic research design and implementation, and genomic health care.[62] In the training category, the program provides institutional research training grants (T32) to support three pre- and post-doctoral training programs in ELSI research, among other support to trainees and early-career investigators.[60] In fiscal year 2020, 18% of the ELSI budget (about $4 million), was spent on the various training programs (J. Boyer, personal communication).

The ELSI Research Program has also invested in activities to build institutional capacity for ELSI research, translate ELSI research, and encourage collaboration among the ELSI scholarly community. For example, in 2004, NHGRI collaborated with the DOE and the Eunice Kennedy Shriver National Institute of Child Health and Human Development (NICHD) to establish the first four CEERs.[61,63] In 2020, it continued the funding for three CEERs (see Table 4) using a limited competition request for proposals (renewal of current CEERs only) with the indication that "NHGRI plans to maintain the CEER program at approximately its current level of funding through FY 2023."[64] In 2019, it funded the Center for ELSI Resources and Analysis (CERA) to build the community ELSI researchers and provide a web-based platform to enhance the production, sharing, and use of ELSI research (for CERA rationale, see Oliver and McGuire,[65] Kaye et al.,[66] and Bell et al.[67]). Although a small portion of the budget, the program has also provided formal support for ELSI studies embedded in large genomics initiatives sponsored by other NHGRI divisions such as the CSER Consortium, eMERGE, the Human Microbiome Project, the Wellcome Trust/NIH H3Africa Initiative, and the NBSeq initiative, as well as supplements to other NIH grants with ELSI components.[60,61,63,68–70]

## ELSI CRITICISMS AND IMPACTS

Over the past 30 years, criticisms have come from scientists, NIH officials, and ELSI researchers themselves. Early on, HGP scientists questioned whether funding the study of consequences was a good use of project funds and worried that an ethics component would signal a need for public scrutiny.[12,40] Because ELSI research shares funding with the science it observes, other commentators have said that ELSI researchers have no choice but to function as translators, mediators, or facilitators of science by manufacturing public acceptability.[71,72] Andrews reported that Watson made a remark at a genetics policy meeting implying that he had configured the ELSI program to minimize its ability to impede HGP progress: "I wanted a group that would talk and talk and never get anything done and if they did do something, I wanted them to get it wrong. I wanted as its head Shirley Temple Black."[49,73] Internal participants and external observers alike viewed the diversion of HGP funds to ELSI as an "unavoidable political tax" that Watson was willing to pay to accomplish his scientific goals.[29] For their part, bioethics scholars worried that ELSI funding would direct the attention of social scientists and humanists away from other pressing issues in biomedicine while at the same time reducing the capacity of bioethicists to critically examine the HGP "either by professionally indebting them to the Project or by redirecting their attention 'downstream' from the Project to its applications."[33]

Scholars in the ELSI community have identified epistemological differences between ELSI researchers, scientists, and clinicians; time pressure; knowledge gaps on both sides; and power imbalances in collaborative relationships as barriers to effective, real-time consideration of ELSI issues.[71,74,75] For thirty years, there has been active discussion over whether ELSI research has had or should have a direct policy impact.[3,29,40,63,76,77] Another criticism is that the focus by ELSI researchers on the "implications of" novel technologies requires its practitioners

> **Box 4. Goals for the ELSI Research Program in the HGP strategic plan for 1998–2003**
>
> ● Examine the issues surrounding the completion of the human DNA sequence and the study of human genetic variation.
> ● Examine issues raised by the integration of genetic technologies and information into health care and public health activities.
> ● Examine issues raised by the integration of knowledge about genomics and gene-environment interactions into nonclinical settings.
> ● Explore ways in which new genetic knowledge may interact with a variety of philosophical, theological, and ethical perspectives.
> ● Explore how socioeconomic factors and concepts of race and ethnicity influence the use, understanding, and interpretation of genetic information, the use of genetic services, and the development of policy.[25]

**Table 3. Summary of the National Human Genome Research Institute extramural research budget, 2018–2021**

| Year | 2018 | 2019 | 2020 | 2021 |
|---|---|---|---|---|
| NHGRI extramural research budget | $391,000,000 | $405,000,000 | $430,000,000 | $437,000,000 |
| ELSI budget | $21,000,000 | $21,000,000 | $22,000,000 | $23,000,000 |
| Percentage of the NHGRI extramural research budget allocated to ELSI | 5.37% | 5.19% | 5.12% | 5.26% |

to use "speculative ethics" and make policy recommendations on the basis of "a possible (and probably inadvertent) exaggerated portrayal of harm" instead of evidence. These commentators raise concern that ELSI findings are conveyed without appropriate nuance and that the resulting "ethics hype" can misinform the public, lead to poor policy decisions, and create backlash against promising research fields.[71,72,78,79]

Despite these criticisms, commentators have credited ELSI research programs throughout the world with the creation of "a healthy culture of skeptical scrutiny" useful for the examination of emerging science and technology.[61,78] More tangibly, ELSI research has been an upstream contributor to several important legislative and judicial outcomes. For example, in 1992, the recommendations of the ELSI Task Force on Genetic Information and Insurance were passed on to the White House Task Force on Health Care Reform, chaired by Hillary Clinton. These recommendations were included in the Health Care Security Act of 1993 and became the public case for health care reform.[33,40] Although the bill failed, the Health Insurance Portability and Accountability Act of 1996 ultimately excluded indications for developing genetic disease as predicted by genetic tests (in the absence of disease) from the list of clauses naming preexisting conditions.[40] ELSI has also been credited with influencing the Congress to extend the Americans With Disabilities Act to offer protection from employment discrimination to individuals with genetic disease or test results predicting the clinical manifestation of future genetic disorders.[40,80] ELSI research findings informed a report by the NIH Secretary's Advisory Committee on Genomics, Health, and Society (SACGHS) that became part of the evidence for the Supreme Court finding against Myriad Genetics, which argued that DNA was excluded from patent eligibility. This important decision contradicted the then generally accepted practice of gene patenting.[61]

ELSI research has also produced several concrete policy outcomes that have positively shaped the conduct of genomic research and protected human rights. Among these are the Genetic Information Nondiscrimination Act of 2008 and the Universal Declaration on the Human Genome and Human Rights.[78,81–84] Other important ELSI accomplishments are improvements to the drafting and ethical review of consent forms for genomic studies; the development of NIH policies for genomic data sharing for the purposes of conducting genome-wide association studies (GWAS); position statements, policies, and recommendations for direct-to-consumer genetic testing; policies, practices, and governance for biobanks and biorepositories at the NIH and other institutions; the adoption of genetic screening guidelines by professional organizations; an executive order protecting federal employees from genetic discrimination in the workplace; analysis and recommendations on returning individual results to research par-

ticipants; and recommendations to end the clinical and research use of race as a biological category.[53,61,78,85–91]

## DISCUSSION

One interpretation of the institutional history of the ELSI program is that the 1996 evaluation ordered by Collins and Patrinos was a deliberate strategy designed to end the ELSI working group, motivated by its ability to set an agenda that was not in alignment with NIH priorities. It could be argued that the group had a direct role in policy formulation because it could operate like an independent commission (as evinced by its task forces on insurance and privacy and draft genetic privacy legislation). Compared with the ELSI working group, the subsequent ELSI advisory groups formulated by NHGRI in its wake, for example, ERPEG (1997–2000), ERA (2000–2012), and GSWG (2012 to the present), could be seen to have a reduced and more internal influence on the NIH scope of work, namely, to analyze the portfolio of ELSI grants and suggest priority areas to the NACHG. Whether this move by Collins was, as Andrews suggested in 1999, an attempt to curtail the "independence" of the ELSI working group is for the reader to decide.[49]

An alternative interpretation is that the 1996 evaluation, even if it had ended the ELSI working group, did not end the production of ELSI-informed science policy. Instead, it catalyzed the transfer of its policy functions to other entities such as the Trans-NIH Bioethics Committee (created in July 1997 by Collins); the Secretary's Advisory Committee on Genetic Testing (SACGT) (chartered by the secretary of health and human services in June 1998), and its successor through 2011, the Secretary's Advisory Committee on Genetics, Health, and Society (SACGHS); and the National Bioethics Advisory Committee (1996–2001), and its successor the President's Bioethics Commission (2001–2009).[57] Although some of these entities were not specifically concerned with genetics policy, SACGHS, which was situated within the NIH Office of Science Policy and was charted to formulate recommendations for the DHHS and other federal agencies on "the range of complex and sensitive medical, ethical, legal, and social issues raised by new technological developments in human genetics," had a remit that closely paralleled the aims of the ELSI working group.[92] In addition, other NIH committees such as the Recombinant DNA Advisory Committee (RAC) (and its successor NexTRAC in the NIH Office of Science Policy) have advised the NIH Office of the Director on ethical, legal, and social issues of emerging technologies beyond genetics. However, few of these entities can be characterized as truly independent of NIH priorities.

Before the ELSI working group and the ELSI grants programs at the NIH and the DOE, ethical oversight of biomedical research in the United States was the remit of independent commissions

**Table 4. Centers of Excellence in ELSI Research funded by the National Human Genome Research Institute through 2024**

| Center of excellence | Focus area(s) | Institution |
|---|---|---|
| Center for the Ethics of Indigenous Genomic Research | Research, education, and outreach for ethical genomic research in partnership with American Indian and Alaska Native communities | University of Oklahoma |
| Genetic Privacy and Identity in Community Settings (GetPreCiSe) | ELSI issues involving genetics, privacy, and identity; related laws and regulatory frameworks; privacy protection technologies | Vanderbilt University Medical Center |
| University of Utah Center of Excellence in ELSI Research (UCEER) | ELSI issues in population-based genetic testing and screening (e.g., newborn screening, prenatal screening, carrier screening, etc.) | University of Utah |

or agency-based advisory panels. These bodies would recommend studies that would then be funded by targeted government contracts or grants.[93] In the initial years of the ELSI extramural grants programs, at least one science policy scholar suggested that a federally chartered commission would have been superior to the ELSI model because the research programs (1) excluded all citizens from participation in setting the policy agenda, except those capable of responding to a grant solicitation, and (2) academicians are "dangerously naive when it comes to public policy."[29] A 1992 report by the House Committee on Government Operations recommended legislation to establish an advisory commission on the ethical, legal, and social implications of the human genome project supported with a portion of the ELSI grant funds because the NIH-DOE ELSI programs are principally designed to support academic research."[43] However, the superiority of commissions (whether NIH dependent or independent) in comparison with the ELSI model did not have universal support.[33,93] Although concerns about the intellectual independence of the field are still present in the minds of ELSI scholars and others today, the decision to establish the HGP as the first federally funded, scientific research program to allocate a portion of its funding to the study of its own impact has been called a "disruptive leap."[93] The "un-commission" (as Eric Juengst described the ELSI Research Program) can be seen as a competitively funded, investigator-initiated research program open to all U.S. scholars, that ensures oversight is near scientific developments on an ongoing basis.[93]

It is our view that the public interest is best served if national science policy is formulated by agencies external to the NIH. However, before the ELSI Research Program at NHGRI, there was virtually no literature (except for narrowly focused technology assessments and sociological studies of technological process) with which to think about the social impacts of a new technology and what action to take. Our interpretation of the ELSI experiment is that to maximize the public benefit of such research, its funders should be clear about assignment of responsibility for research translation activities. We agree with Burke and colleagues, who conclude, on the basis of their experiences with the "translational imperative" of the CEER program, that it would be a mistake to require that ELSI research programs demonstrate a direct impact on science or health policy.[63] However, for the ELSI Research Program to have maximum utility, it should facilitate the translation of ELSI scholarship. To that end, we would suggest that it evaluate the appropriateness of requiring academicians to produce policy-relevant grant prod-

ucts, and either discontinue this legacy requirement for grantees (and arrange for this responsibility to belong to well-informed others) or, when it is appropriate, train and firmly support a set of investigators from policy-focused disciplines to create specific products. It could also create pathways for ELSI scholars to engage with a broad range of stakeholders.

It has been asserted (as a criticism) that the ELSI program is principally designed to support the production of academic research.[29] If this is so, what challenges does its potential entanglement with the strategic priorities of the NHGRI pose for the classification of ELSI as a field of intellectual inquiry? Its strategic plans suggest that NHGRI has been supporting basic science research on the human genome for the purpose of its eventual clinical translation and use.[25,58,94] The achievement of this goal will ultimately require the recruitment of thousands of subjects to genomic research, the adoption of genomic technologies in the clinic, and their acceptance by patients. The singular focus suggested by this project may partly explain the gap in the NHGRI ELSI portfolio identified by the ERPEG in 2000, namely, a lack of studies that explored the broader social implications of enhanced knowledge of human genetics and genomics. Work that has no translational potential may not be prioritized for addition to the portfolio. Although out of scope for the present paper, an updated portfolio review may identify evidence that many funded projects are designed to recruit members of the public to the NHGRI project. It may find additional gaps, especially in areas that may be critical of genome science.

There are also questions that may be beyond the scope of NHGRI ELSI Research Program grantees. For example, the suggestion by Fabi and Goldberg that a focus by bioethics funders on "genetics, genomics, neuroethics, and the ethics of other emerging technologies disproportionately harms People of Color."[95] They argued for a more just allocation of research dollars to the field of bioethics because "a narrow focus on emerging technologies, such as genetic and genomic technologies, reflects a priority set that does not always represent the needs of all sectors of society."[95] Internal and external evaluations have suggested that the ELSI Research Program has struggled to achieve demographic diversity in its recruitment of new investigators.[52,57,95] The allocation of research funding to ELSI, especially if the ELSI program continues to struggle in this way, could limit the kinds of research questions that are explored as well as contribute to the existing, unequal allocation of NIH funding to scholars of color.[95–97] Perhaps the outcome here depends upon the results of the NIH anti-racism initiative, UNITE,

announced in March 2021, which aims to reduce barriers to achieving racial equity in the NIH-supported and external scientific workforce (including ELSI).[98] It also depends upon the actions of other institutions in the bioethics ecosystem.[95]

The past three decades of sustained investment by NHGRI have created a substantial body of scholarship and successive cohorts of trained ELSI practitioners. These investments enable emerging ELSI issues to be described with greater specificity and approached with accumulated knowledge and experience. Our experience will be of great value in the coming years as ELSI scholars find themselves engaged in a study of genome science that "has become increasingly woven into the fabric of biomedical research, medical practice, and society."[64] As the reach of genome science stretches beyond its laboratory origins, it is more imperative that ELSI strive to be an independent field of inquiry driven by the needs and concerns of those who are affected by developments in genome science. If NHGRI and genome science are ready to seriously examine the social and environmental conditions that interact with genetic risk in the production of human disease, the urgent need to engage diverse communities, and the varied personal and cultural influences on the interpretation and use of genetic information, it should look to ELSI researchers, who now can productively draw on thirty years of experience, to make meaningful contributions to the analysis of those issues in the fourth decade of ELSI study.

## SUPPLEMENTAL INFORMATION

## ACKNOWLEDGMENTS

We would like to thank Eric Juengst and the anonymous reviewers for their comments, Joy Boyer for organizational charts and budget data, and Caroline B. Moore for reference formatting and tables. This essay was supported by the National Human Genome Research Institute at the National Institutes of Health under award U24HG010733. The content is solely the responsibility of the authors and does not represent the official views of the National Institutes of Health.

## AUTHOR CONTRIBUTIONS

Conceptualization, D.D.D., S.S.-J.L., and M.K.C.; formal analysis, D.D.D.; investigation, D.D.D.; writing – original draft, D.D.D.; writing – review & editing, D.D.D., S.S.-J.L., and M.K.C.; funding acquisition, S.S.-J.L. and M.K.C.; supervision, M.K.C.; visualization, D.D.D.; project administration, D.D.D.

## DECLARATION OF INTERESTS

All authors declare salary support from the ELSI Research Program, National Human Genome Research Institute, National Institutes of Health. S.S.-J.L. was a member of the Genomics and Society Working Group from 2016 to 2020 and currently serves on the National Academies of Science, Engineering and Medicine Committee on Use of Race, Ethnicity, and Ancestry as Population Descriptors in Genomics Research.

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
