## [Document S1. Transparent peer review records for Dolan et al · Cell Genomics]

Cell Genomics, Volume 2

Supplemental information

**Three decades of ethical, legal,
and social implications research:**

Looking back to chart a path forward

Deanne Dunbar Dolan, Sandra Soo-Jin Lee, and Mildred K. Cho

Title

Author list: Deanne Dunbar Dolan, Sandra Soo-Jin Lee, Mildred K. Cho

Summary

Initial submission: Received : **Mar 25, 2021**

Scientific editors: Orli Bahcall, Rosalind Mott, Laura M. Zahn

First round of review: Number of reviewers: 2
Revision invited : Sep 9 2021
Revision received : Feb 23 2022

Second round of review: Number of reviewers: 1
Accepted :

Data freely available: N/A

Code freely available: N/A

This transparent peer review record is not systematically proofread, type-set, or edited. Special characters, formatting, and equations may fail to render properly. Standard procedural text within the editor's letters has been deleted for the sake of brevity, but all official correspondence specific to the manuscript has been preserved.

Referees' reports, first round of review

Reviewer 1:

Tremendously difficult to summarize 30 years of ELSI research and politics without taking "sides." Overall, I thought you did a reasonable job, but I would push you further to look more deeply at three issues: (1) the original agenda from the perspective of James Watson and also the ELSI working group; (2) the total control taken by Francis Collins of the ELSI Working Group after Lori Andrews resigned as its chairperson (recounted in her book, *The Clone Age*), and the simultaneous movement of "public policy" away from suggesting specific laws and regulations, especially in the arena of genetic privacy; and (3) the intellectual and practical relationship between ELSI researchers and the National Academy of Medicine reports on genetics (including gene editing).

Some specifics: (1) Of some historical and intellectual interest, the first extramural workshop funded by NIH was held in Bethesda in January 1991. Its purpose was to develop a prioritized social policy research agenda for the Human Genome Project. The four areas the group suggested, in rank order, were: 1. When and how should new genetic tests be introduced into medical practice?; How can confidentiality and privacy of an individual's genetic information be preserved?; How can genetic discrimination by employers and insurance companies be prevented?; and 4. How might the Human Genome Project affect our concepts of 'disease,' 'normalcy,' and 'humanness.'? Annas, GJ & Elias S, eds, *Gene Mapping: Using Law & Ethics as Guides*, Oxford U. Press, 1992. Although not stated in these exact terms, these four items remain on the top of any ELSI research priorities to this day. (2) The resignation of Lori Andrews as chair of the ELSI working group (which you note) is of much more influence over the ELSI agenda than you give it credit for. It marked the end of any independence of the ELSI research agenda, removing the Working Group's independence (such as it was), and putting complete control in the hands of Francis Collins (who had by then replaced James Watson). In retrospect, Prof. Andrews suggests that ELSI may not have been seen as all that important by Watson either. In her words, "At a genetics policy meeting I learned why James Watson had formed the ELSI Working Group. Watson implied that the ELSI Working Group had been created not to set ethical standards but to let the science proceed unimpeded. 'I wanted a group that would talk and talk and never get anything done,' Watson said, 'and if they did do something, I wanted them to get it wrong.' 'I wanted as its head Shirley Temple Black.'" Lori Andrews, *The Clone Age*, Henry Holt, 1999 at 206. (3) The question of whether ELSI should get directly into making policy is a good one, but one cut short by the resignation of Lori Andrews (over a related issue). With funding from the DOE's ELSI project, researchers responded to the request to draft a federal genetic privacy act. This project was completed in February 1995 (Annas, Glantz & Roche, *The Genetic Privacy Act and Commentary* (1995)). The act was presented to a meeting of the ELSI Working Group in 1994, and got unanimous "approval" at the meeting. Thereafter, researchers (especially, but not only, pathologists) began a campaign against the draft (worried especially that giving individuals an ownership interest in their DNA could make genetic research harder to do), which in the end persuaded Francis Collins not to support this bill. The experience, coupled with the resignation of Lori Andrews at about

the same time, seemed to make ELSI far less likely to engage in legislative drafting in the realm of genomics. (There is some irony that concerns about genetic privacy have more recently taken a front seat in the "All of us" project. As you put it (correctly): "The 1996 ELSI Evaluation Committee recommended the exclusion of direct policy formulation from the remit of the ELSI program." This helps explain why.

I would suggest that you move the material on "Global ELSI Initiatives" to an appendix because as a list without any attempt at synthesis or analysis it doesn't add enough to your paper to warrant the space it is given.

Finally, I'm not convinced that your survey of the ELSI literature leads to your final conclusion, and I ask that you reconsider it.

Reviewer 2:

This is an excellent review of the history of the ELSI program, and builds on the previous efforts (McEwen, Juengst, etc.). It is clearly written and timely. It should be published, and can be an input to NHGRI (and NIH more widely) thinking through where it wants to go with the ELSI research program, now that the "pause" in the CEER program is coming to an end, and NIH's budget it slated for renewed growth.

This piece is stronger on descriptive history than analysis of the lessons learned. I have very few comments about the history, except to suggest that they add a citation to Lori Andrews's chapter from her book "The Clone Age" that focuses on the aspects of the ELSI program as PR front for genomics science, a "shield" to insulate the science from criticism so the science can proceed apace. Jim Watson gave her an unguarded interview, and this was indeed one of his intents, but he was also sincerely interested in "this is not just for scientists to decide," and he was no doubt showcasing the provocateur self when he talked about the purpose of the program. But it's probably the best and sharpest such critique of the program. Lori was still smarting from resigning as ELSI Working Group chair five years earlier, and the chapter is not fully balanced. I don't have access to my copy, but there is a chapter that focuses on this issue that is sharper than the Juengst and other critiques cited.

The article is well worth publishing for the update of the history, and also adding some details to that history, no doubt benefiting from Chris Donahue's historical work at NHGRI.

But its main use will be to guide decisions in the future, and as an analytical piece about "lessons learned" it could be strengthened to good effect. The goals of the program are diverse. Most of the items on the 1989 initial Working Group list and the revamped goals after Francis came to NCHGR are oriented to helping identify issues (early warning and the initial conferences to orient the field), and then to study issues that were cropping up in the science and application (the early work on genetic testing for CF, breast cancer, Alzheimer's, Huntington's, etc.). Some projects have been deep "embedded" projects (EMERGE, CSER, ClinGen have ELSI components, for example), and this "embed" notion has spread to some extent to All of Us and BRAIN Initiatives. But it's striking the degree to which it is **not** been adopted by other institutes even at NIH, and there's little discussion of how and why the DOE program dissipated, despite the very strong Congressional signal that DOE needed to have an ELSI component to its science. And where the "parallel research program" for ELSI has been adopted (e.g.,

the US nanotech, or the Canadian GenomeCanada GE3LS), sometimes each major project is expected to have an ELSI component, and sometimes ELSI research is separately and independently funded. What are the strengths and weaknesses of the "embed" versus "independent" modes? This is crucial to NHGRI's internal debate about the role of CEER centers, and this article is an opportunity to signal the community how NHGRI is thinking about that central question.

As one niggler with the history. The Walters and Rothstein evaluations of the program were partly a reaction to Nancy Wexler, Lori Andrews and Troy Duster succession of working group leadership, but those resignations were symptoms as much as causes. The underlying tension was whether Francis and Kathy Hudson would control the agenda, or an external group. And the role of the working group in formulating policy was hotly contested, and indeed the main bone of contention. The evaluations did indeed caution against formulating policy within an NIH advisory committee (which is what the working group was—or should have been explicitly framed to be, a FACA committee). But that is a clear divergence from the expectations in Congress, and the initial goals (see #4 of the 1989/1990 "develop policy options" that Leslie Fink included in her summary). The real frustration of folks like Kathy and Francis trying to get things done in a highly political environment and the independence of an advisory committee and a research program that was not tightly harnessed to NCHGR/NHGRI policy priorities but left to investigators and study sections does not really come through. And the decision to remove the "policy options" function was a judgement call, and one that was (and should have been) hotly contested. Those evaluation reports were crucial in shifting the main power to set the ELSI research agenda to internal NHGRI documents and staff, and to reduce the independence and power of the working group as a national bioethics committee (and it was serving both functions, to some degree—witness the ELSI working group task forces on insurance and on genetic testing). One interpretation is that the evaluations were intended to rein in the ELSI Working Group from those policy-oriented activities, a goal that was largely accomplished, as the result is a group of ELSI Research Advisors who plug into the NIH-standard advisory council. That group has much lower profile, focused on the ELSI research agenda, and shed the "national bioethics forum" aspirations. That certainly simplified NHGRI politics by reducing the political salience of the ELSI working group? But was what was good for NHGRI management good for the country? I don't know the answer to that, and probably no one does, but it was a fateful choice that should be explicitly addressed. The framing of the tasks for those evaluation committees was crucially determined by Kathy and Francis. That history is pretty well known to those who lived through it, and the current account seems bowdlerized.

The ELSI program is housed in a research institute that fosters some very basic research. Some of the ELSI grants were intended to roughly parallel this model in supporting purely intellectual contributions to advance the field—that is, unfettered investigator-initiated research. How much? How have those fared. The list of problems that ELSI research has helped to address is relevant to the "problem solving" and "early warning" goals, which stem from a more practical rationale for the research (more "applied" in Vannevar Bush's terms). How much of the ELSI portfolio is devoted to that?

In general, the piece would be greatly strengthened if it included some evaluation, specified the different goals explicitly, included some funding and portfolio-analysis data that is expected of any NIH program evaluation, and some "strengths and weaknesses" assessment of how the program has achieved (or not) the goals. The history is there, now flesh it out with some funding data, e.g., fraction of

ELSI relative to NHGRI budget overall, fraction devoted to CEERs, fraction devoted to "intellectual advancement" versus "problem solving" versus "field support (e.g., the CERA and T32s) versus "embedded ELSI" in major tech initiatives. These overlap, so the analysis cannot be simple and completely clean, but these are the real issues facing the field, and these problems of classification and analysis plague the scientific programs at NIH just as much as ELSI. And yet NIH has a history of program reviews that matter.

The main value of this article will be to review and update the history of the ELSI program. But one important purpose of that history is to inform the choices facing NHGRI for the program's future, and to address why the model has remained largely restricted to NHGRI. Three decades ago, OTA and the Institute of Medicine recommended that the ADAMHA institutes have ELSI programs. They never really developed them. Why? Imagine, if you will, if NIAID had an ELSI component (beyond strong ties to the Clinical Center bioethics program) to complement its vaccine-development effort, and how it might have anticipated the issues that have proven to be major obstacles to pandemic management (and those obstacles have largely been social, political, and economic, not just science and technology). Or how NIH is grappling with AI and infotech, with the usual highly decentralized institute-and-Center responses that are somewhat coordinated, but largely bereft of the kind of social science research that could have been going on for the past decade if NIH had a capacity for exploring ELSI of biomedical research generally, analogous to NHGRI's capacity for ELSI of genomics.

This critique may seem harsh and add too much work, but the NHGRI ELSI program really is the largest and longest-standing such program in the world, and now decisions are being made about its future configuration. This article is an opportunity to do the hard work of thinking through what that might look like, based on historical *assessment* not just historical description (which this piece does a really nice job of). I would encourage the author to do some of that hard work. Perhaps not in this piece (as it's already fairly long), but perhaps by alluding to (and pledging to write) a future analysis. But if that is the solution chosen, the current history is still incomplete without the budget data and specification of goals and approaches noted above. A paragraph that lays out the "portfolio" of ELSI projects with different goals, and the budget breakdowns used to guide current NHGRI decisions should be laid out. And one result might be to refine those categories and clarify the disparate goals and adjust the metrics to accommodate the diverse goals of different projects. (As one example, the "innovation" criterion for grants is highly relevant for projects funded to advance ELSI as a field; but it is actually counterproductive for most "embedded" projects where the purpose is to help channel evidence about what's known to project leaders and anticipate issues using standard methods that are tried-and-true, because there the purpose of ELSI is to improve the science, not advance ELSI as a field—except as "embeddedness" can itself be subject to some innovation, but it really is not the main purpose.)

Finally, there's an intellectual incoherence embedded in the final few pages. Sometimes ELSI is described as a field. It's clearly not a discipline, but rather something else. And yet the CEERs once had training components and there are ELSI T32 training programs. By those criteria, it is a "field" in the sense of having a community of folks who read one another's work and perform Merton's CUDOS functions for that community. But it's not a field in some other senses, as it is sometimes helping solve problems, sometimes forecasting, and sometimes it really is trying to advance intellectual contributions to existing

fields, disciplines, or professions (e.g., clinical practice, patent law, philosophy of biology, moral philosophy). Is it a function (or functions) or a field? Or both?

Authors' response to the first round of review

Dear Dr. Bahcall,

Thank you for a thorough peer-review process and your editorial comments on our March 2021 submission, A history and actionable priorities for the fourth decade of ELSI research. We have substantially revised the work to provide a more balanced account of the ELSI Research Program at the National Human Genome Research Institute (NHGRI). Specifically, the following reviewer and journal concerns are fully addressed in the revised manuscript:

1. We clarified our claim that ELSI research is a field of study.
2. We described the 1991 workshop that Reviewer 1 felt identified 4 areas of social policy for the HGP and remain the ELSI research priorities today.
3. We moved lists to boxes or tables and have added a table that details the portion of the NHGRI extramural research budget allocated to ELSI over the past 4 years.
4. In order to better characterize the strengths and weaknesses of the ELSI Research Program, we discuss published evaluations by science policy or ethics scholars as well as in-depth, formal evaluations by advisory panels appointed by NHGRI (e.g., Hanna, 1995; Juengst, 1996; Rothstein et al., 1996; Walters et al., 2000; ELSI Research Advisors and ELSI Policy and Planning Group, 2003; and ELSI Research Advisors (ERA), 2005).
5. Given the prominence of investigator-initiated research in the NHGRI ELSI Program portfolio, the content of which is informed both by study sections (peer review) and Council priorities, we were hesitant to follow Lori Anderson's lead and suggest that the NHGRI Director can exercise total control over the ELSI research agenda. We also felt the suggestion by Reviewer 2 that the 1996 external evaluation of the ELSI Working Group and the extramural grants programs at the NIH and the DOE ordered by Collins and Patrinos marked the end of independence for the ELSI research agenda ("Those evaluation reports were crucial in shifting the main power to set the ELSI research agenda to internal NHGRI documents and staff, and to reduce the independence and power of the working group as a national bioethics committee") was somewhat mitigated by the subsequent formation of two national committees focused on the examination of ELSI issues. Nonetheless, we acknowledge this as a possible interpretation of the program history in the new conclusion.
6. The new conclusion provides an interpretation of the program history and recommendations to NHGRI on the future direction of the funding program and the field of study it supports.

Although our revision was substantial, a few remaining issues deserve comment here:

Response to Reviewers

1. We did not include funding or portfolio analysis data for each Center for Excellence in ELSI Research (CEER) because we felt a detailed analysis of CEER budgets, which are only a subset of the overall NIH budget for ELSI research, would be both outside of the current scope and unlikely to contribute meaningfully to the paper.

2. We did not find a published rationale for the sunset of the NHGRI CEER program or the Department of Energy (DOE) ELSI Branch. In the later case, we suspect (but were unable to document) the extramural ELSI funding program run by the DOE was phased out in 2003 with the completion of the HGP. However, the revised paper mentions the CEER sunset, includes detail about the funding differential between the NIH (2/3) and the DOE (1/3) in the HGP, provides a reference for the DOE rationale for joining the HGP (interest in energy-related damage to the human genome), and explains why the DOE funded its ELSI program.

3. We have updated the manuscript to include a breakdown of the NHGRI budget that illuminates the portion spent on investigator-initiated research vs. other areas (training, embedded ELSI, etc). However, the budget data we obtained from NHGRI program staff do not offer the granularity that would be required to divide these expenditures into “purely intellectual” and “applied” categories, as suggested by Reviewer 2. A future analysis that sought to classify individual grants into these categories could be conducted, but we felt the distinction may be too binary and the analysis potentially unproductive.

4. Although we tend to agree with the suggestion by Reviewer 1 that both Andrew’s resignation and a campaign by researchers that persuaded Francis Collins not to support the genetic privacy legislation drafted by George Annas and others with DOE ELSI funding, explains the shifting definition of a “public policy” role for ELSI away from specific laws and regulations, we did not find published support for this idea. Instead, we describe formal, internal evaluations by panels of experts that recommend the relocation of the ELSI Working Group’s policy formulation function to other NIH policy offices. Although we were unable to document the Reviewer’s perspective, we acknowledge this possible interpretation of the program history and make our own argument about the optimal location of science policymaking in the new conclusion.

5. We did not take up the discussion, as suggested by Reviewer 2, of the implications of “embedded” ELSI for the intellectual independence of the field of study. This issue has been discussed in detail elsewhere, e.g., Conley et al., 2020; Joly et al., 2016; Balmer et al., 2015; and Seltzer et al., 2011). Further, at only 3% of the ELSI Research Program budget in 2020, we did not feel that this discussion was especially timely.

6. Reviewer 2 asks that we address why, in their view, the ELSI model has remained largely restricted to NHGRI. While we would love to be in a position to take on this question, we expect that, given the many actors involved (including the U.S. Congress) we would likely find many answers and perhaps none that are directly or definitively motivated by qualities, success, or failures of the ELSI Research Program. As we explain in the manuscript, the ELSI extamural grants programs were the product of a broad set of cultural currents that were unique to the late 1980s. Further, the Reviewer’s contention contradicts our presentation in Table 2 which depicts the broad global adoption of funding set-asides dedicated to ELSI research within national scientific initiatives.

Despite the above limitations and its unusual length compared to a typical Perspectives manuscript, we truly hope the revised manuscript is satisfactory to the reviewers and the journal. Thank you for your further consideration of this work.

Best wishes,

Deanne Dolan, PhD
Stanford University

Referees' report, second round of review

Reviewer 1:

The subject matter is worth the research effort, and your changes were on the whole responsive to the reviewer comments. A few things I would ask you to consider (or re-consider).

1. Although you do cite her main book on the subject, you treat Lori Andrews, a central participant in the debate over governance of the ELSI program as someone you don't seem to trust as an honest reporter. I don't usually suggest that authors be more careful of "tone" but you should in regards to Andrews at least (see, especially where you introduce Andrews as simply "a lawyer" [rather than as a law professor and legal scholar; see eg., Jorge Contreras, *The Genome Defense: Inside the Epic Legal Battle to Determine who Owns Your DNA* (2021) which recounts the central role Prof. Andrews had in the litigation that eventually led to the Supreme Court decision on gene patenting]; and then when you cite Andrews for important detailed you use the phrase "Andrews claims" (twice on p. 6 alone), and also the phrase "according to Andrews" None of the material attributed to her has been questioned--and if you do not believe her you should say why, not engage in what looks like marginalizing her conclusions based on living through the events. [likewise, at p. 10, "according" to Andrews ...Watson "allegedly"--unlike most of Watson's racist and sexist remarks, this one was at a public event and witnessed by many others, and never denied by Watson. at the least, the "allegedly" should be removed.

2. The proper citation of the "Genetic Privacy Act" (referenced at p. 8) is George J. Annas, Leonard H. Glantz & Patricia A. Roche, *The Genetic Privacy Act and Commentary* (1995). (easily accessible on the net) The "draft" of this report was presented to the ELSI Working Group at an in person meeting (by Annas) in 1994 and endorsed by the group unanimously, including Francis Collins (who later changed his mind). It may (or may not) be worth noting that virtually all of the consent and privacy provisions of the Genetic Privacy Act were adopted by the "All of Us" project.

3. p. 12, describing the post-Andrews ELSI program as "a public relations stunt" is probably too strong, but there is no doubt that Collins saw genome-related policy making potentially moving out of his office and he moved to contain and control it.(pretty standard Washington behavior)

4. on p. 13: the National Academy of Sciences doesn't really work the way you describe it.

5. Good points (I think) on pages 14 and 15.

6. Its a bit concerning that the only person you acknowledge personally is Eric Juengst who, of course, has his own view (and, for example, who co-authored the foreword to the Gene Mapping book [the source of the research priorities now summarized in box 3] with James Watson.

7. I know its arguably beyond the scope of your paper, but I (and I'm sure many readers) would be very interested in your views on the efforts DARPA has put into developing its own ELSI program (especially in it unclassified Safe Genes project) which, on the surface at least, is homage to the ELSI model.

Authors' response to the second round of review

Dear Dr. Mott,

Thank you for your work as handling editor for our paper. We very much appreciate your support for the expanded Perspective word limit, extra figures/boxes, and additional references. We have reduced the manuscript length from 12,000 to 10,479 words and removed the STAR Methods section, which is not required for the Perspective format. Please find our point-by-point response to the reviewer comments below.

Reviewers' Comments:

Reviewer #1: Subject matter is worth the research effort, and your changes were on the whole responsive to the reviewer comments. A few things I would ask you to consider (or re-consider).

1. Although you do cite her main book on the subject, you treat Lori Andrews, a central participant in the debate over governance of the ELSI program as someone you don't seem to trust as an honest reporter. I don't usually suggest that authors be more careful of "tone" but you should in regards to Andrews at least (see, especially where you introduce Andrews as simply "a lawyer" [rather than as a law professor and legal scholar; see eg., Jorge Contreras, *The Genome Defense: Inside the Epic Legal Battle to Determine who Owns Your DNA* (2021) which recounts the central role Prof. Andrews had in the litigation that eventually led to the Supreme Court decision on gene patenting]; and then when you cite Andrews for important detailed you use the phrase "Andrews claims" (twice on p. 6 alone), and also the phrase "according to Andrews" None of the material attributed to her has been questioned--and if you do not believe her you should say why, not engage in what looks like marginalizing her conclusions based on living through the events. [likewise, at p. 10, "according" to Andrews ...Watson "allegedly"--unlike most of Watson's racist and sexist remarks, this one was at a public event and witnessed by many others, and never denied by Watson. at the least, the "allegedly" should be removed.

We thank the Reviewer for these comments and have made the suggested corrections.

Our intention was never to disrespect Lori Andrews or her contributions. In round 1, Reviewer #2 cautioned that the relevant chapter in the Clone Age was not fully balanced. Given this, we searched for corroborating accounts of these events in the literature and did not find any. As the Reviewer notes, we also did not find any denial of this account by Watson.

2. The proper citation of the "Genetic Privacy Act" (referenced at p. 8) is George J. Annas, Leonard H. Glantz & Patricia A. Roche, *The Genetic Privacy Act and Response to Reviewers Commentary* (1995). (easily accessible on the net) The "draft" of this report was presented to the ELSI Working Group at an in person meeting (by Annas) in 1994 and endorsed by the group unanimously, including Francis Collins (who later changed his mind). It may (or may not) be worth noting that virtually all of the consent and privacy provisions of the Genetic Privacy Act were adopted by the "All of Us" project.

We have made this correction. Given we are well over the journal word count, we will thank the reviewer for their insights about the All of Us project and save these details for another paper.

3. p. 12, describing the post-Andrews ELSI program as "a public relations stunt" is probably too strong, but there is no doubt that Collins saw genome-related policy making potentially moving out of his office and he moved to contain and control it.(pretty standard Washington behavior)

We have removed this language.

4. on p. 13: the National Academy of Sciences doesn't really work the way you describe it.

We reviewed our understanding of the cited paper and removed our mention of the NAS after determining that it was not a relevant detail.

5. Good points (I think) on pages 14 and 15.

We thank the reviewer for this comment.

6. Its a bit concerning that the only person you acknowledge personally is Eric Juengst who, of course, has his own view (and, for example, who co-authored the foreword to the Gene Mapping book [the source of the research priorities now summarized in box 3] with James Watson.

We acknowledge Eric Juengst because he read and commented on an earlier version of the manuscript. The reviewer is correct, Juengst co-authored the foreword of the 1992 book, Gene Mapping: Using Law & Ethics as Guides. We suspect this is because, in 1992, James Watson was the Director of the National Center for Human Genome Research (NCHGR) at the NIH and Eric Juengst was the head of the NCHGR ELSI Branch. In round 1, Reviewer #1 asked us to include the social policy research agenda that appeared in Gene Mapping and we did so. We are unaware of any relationship, past or present, between Juengst and Watson that would create a conflict of interest for us.

7. I know its arguably beyond the scope of your paper, but I (and I'm sure many readers) would be very interested in your views on the efforts DARPA has put into developing its own ELSI program (especially in it unclassified Safe Genes project) which, on the surface at least, is homage to the ELSI model.

We thank the reviewer for this comment and agree that this is beyond the scope of the present manuscript.

We hope these responses and the revised manuscript are satisfactory to the reviewers and the journal. Thank you for your further consideration of this work.

Best wishes,

Deanne Dolan, PhD

Stanford University